# Analysis of Green Development of Aquaculture in China Based on Entropy Method

**Xing Ying and Ping Ying ***

School of Economics and Management, Shanghai Ocean University, Shanghai 201306, China
* Correspondence: yping@shou.edu.cn; Tel.: +86-156-9216-5517

**Abstract:** China is the world's largest producer and consumer of aquatic products. With the rapid economic development of our country and the great improvement of people's living standards, people have put forward higher requirements for beautiful water ecological environments and high-quality aquaculture products, and promoting aquaculture green development has become an important initiative to promote high-quality sustainable development of aquaculture. Therefore, the assessment and analysis of the current green development level of the aquaculture industry have important practical significance for comprehensively grasping the current green development status of the aquaculture industry in China, purposefully breaking the development bottleneck, and promoting green and sustainable development of the aquaculture industry. This paper obtains comprehensive evaluation results of the green development of aquaculture in China from 2012 to 2021 by constructing a green development index system of aquaculture and applying the entropy method. The results show that the green development level of aquaculture in China has a fluctuating upward trend, with a good development momentum. The production and living standards of fishermen are important aspects that affect its development. In order to further promote the green development of aquaculture in China, countermeasures and suggestions are put forward from the following aspects: improving the green subsidy policy, improving the level of supervision and service, strengthening scientific and technological innovation and the transformation of scientific and technological achievements, and making full use of the unique advantages according to local conditions.

**Keywords:** aquaculture; green development; evaluation index system; entropy method

## 1. Introduction

China is the world's largest producer and consumer of aquatic products, and its production of cultured aquatic products has increased 99.69-fold from 524,000 tons in 1949 to 52.242 million tons. However, the intensive development of aquaculture activities over the years, the disorganized expansion of the scale-up of production, the blind pursuit of high-density and high-volume production, and the achievement of aquaculture growth through unsustainable exploitation of aquatic resources have led to the gradual and increasing frequency of eco-environmental pollution of cultured waters and animal diseases in farming [1,2]. With the rapid development of the national economy, the relationship between supply and demand of aquatic products tends to be balanced, and there is a change in people's consumption attitudes [3], where more and more people realize the important role of aquatic products for health and nutrition balance. People put forward higher requirements for beautiful water ecological environments and high-quality aquaculture products. In order to promote the sustainable and high-quality development of aquaculture, in January 2019, with the consent of the State Council, ten ministries and commissions including the Ministry of Agriculture and Rural Affairs jointly issued the Opinions on Accelerating the Green Development of Aquaculture Industry [4], which made a comprehensive deployment around strengthening the scientific layout, changing aquaculture methods, improving aquaculture environments, strengthening production supervision, expanding

development spaces, strengthening policy support, and implementing safeguard measures. The release of this document signifies that the transformation and upgrading of China's aquaculture industry have entered a new stage of development [5]. Exploring healthy and green aquaculture of aquatic products and guiding the industry from pursuing "quantity" growth to pursuing "quality" improvement have become new topics for policymakers and researchers [6]. Promoting the green development of aquaculture has become an important measure to deal with the contradiction between the consumption upgrading of aquatic products and traditional aquaculture production and promote the sustainable development of aquaculture. Therefore, it is of great practical significance to scientifically and objectively evaluate the current green development level of aquaculture in China and comprehensively grasp the status quo of the green development of aquaculture in various regions to break through the bottleneck of development in various regions and promote the formulation and implementation of relevant policies.

In recent years, scholars, policy, and industry participants have gradually started research on green aquaculture to achieve the dual objectives of product safety and quality and environmental sustainability. There are different definitions and connotations of the green development of aquaculture in academic circles. Lu C C [7] believes that the green development of the aquaculture industry is constrained by the ecological capacity and resource-carrying capacity of water areas, relying on advanced management concepts, science and technology, and material equipment to form a new development model with efficient resource utilization, a stable ecosystem, a good production environment, and safe product quality. Yue D D and Wang L M [8] defined it as a new mechanism to achieve environmental friendliness, technical efficiency, product safety, fisherman's income increase, and consumer satisfaction through technological and institutional innovation with the goal of harmonious coexistence between human aquaculture activities and the natural environment. Cao J H [9] believes that the green development of the aquaculture industry is a form of industrial development with the premise of water ecological environment capacity and resource-carrying capacity and the goal of sustainable improvement of human well-being and ecological well-being. Zou L L [10] pointed out that, unlike the development of aquaculture at the cost of the environment in the past, in view of green growth, it is necessary to give priority to the development of aquaculture. Green growth attaches importance to both environmental protection and high productivity. Liu Y Y [11] proposed that the innovation and adoption of new technologies are important ways to achieve sustainable development of aquaculture. Chen J Y [12] believes that the basic connotation of the green development of aquaculture includes protecting fishery resources and the ecological environment, innovating fishery technology, ensuring the quality and safety of aquatic products, and standardizing management according to law. In terms of evaluation methods, Yang Z Y [13] et al. constructed a green development index from three aspects of industrial economic development, production input, and environmental pollution output and measured the green development level of China's marine aquaculture industry by using methods such as the super-efficiency non-radial DEA model. Li X F and Jiang Q J [14] constructed an indicator system for the green development of the aquaculture industry from four aspects—aquaculture resources subsystem, aquaculture economy subsystem, aquaculture environment subsystem, and aquaculture society subsystem—and used the principal component analysis method to evaluate the green development level of the aquaculture industry in the Yangtze River Economic Belt. Yue D D [15] et al. constructed an evaluation index system for the green development of marine aquaculture from the four dimensions of resource conservation, space expansion, environmental friendliness, and product greenness.

At present, most of the research on the green development of aquaculture is qualitative research, focusing on the connotation and influencing factors of the green development of aquaculture, and there are few quantitative studies on the green development of aquaculture. Based on the existing research, this paper sorts out the definition and connotation of the green development of aquaculture, further constructs the index evaluation system,

and uses the entropy method to measure the level of green development of aquaculture. The entropy method is an objective evaluation method that has been widely used in many research fields and has achieved certain results. Zhao H. J. [16] et al. used the entropy method to evaluate the level of green agricultural development based on panel data from 13 major grain-producing provinces in China from 2003 to 2017. Shen, Z. [17] et al. used the entropy value method to analyze the development of green transportation in Zhoushan City in 2016 and 2018. Shang Z. J. [18] et al. used the entropy value method to determine the index weight, calculated the comprehensive development index based on provincial panel data, and evaluated the governance performance of public cultural services. Because the entropy method can determine the importance of the index according to the degree of dispersion of the data itself, it has the characteristics of objectivity and accuracy, so it has been favored by many scholars and widely used in evaluation research. In this paper, the entropy weight model will be introduced to analyze the green development trend and current development level of aquaculture in China. The problems existing in the green development of aquaculture in China and the important factors affecting its development can be further discovered. In addition, relevant policy suggestions are put forward in view of the problems shown so far, hoping to break through the bottleneck of development in a targeted manner so as to promote the sustainable development of aquaculture in China and enrich the research on the green development of aquaculture.

The innovations of this paper are as follows:

First of all, the definition and connotation of green aquaculture development in existing studies have not been clearly defined. By combing the existing literature, this paper gives a new view on the definition and connotation of green aquaculture development. On this basis, the evaluation system of green development indicators of the aquaculture industry was constructed to comprehensively evaluate the green development of aquaculture in China.

Secondly, this paper takes the green development level of aquaculture in China as the research object, expands the scope of research, and grasps the national aquaculture situation in a macro manner. In terms of research methods, the entropy method was used to measure the level of aquaculture green development, and more objective evaluation results were obtained, which enriched the research on aquaculture green development.

## 2. Materials and Methods

### 2.1. Connotation and Definition of Green Development of Aquaculture

At present, the connotation of the green development of aquaculture has not been clearly defined, but most scholars have given views on the ecological environment, product safety, technical efficiency, and economic benefits. In January 2019, ten ministries and commissions jointly issued the "Opinions on Accelerating the Green Development of the Aquaculture Industry", which put forward new requirements for the transformation and upgrading of China's aquaculture industry, and the opinions made comprehensive arrangements around strengthening the scientific layout, transforming breeding methods, improving the breeding environment, strengthening production supervision, broadening development spaces, strengthening policy support, and implementing safeguard measures. Drawing on existing research, this study believes that the connotation of the green development of the aquaculture industry is mainly reflected in two aspects: the breeding process and aquaculture products. The greening of the aquaculture process is reflected in the continuous improvement of aquaculture technology by aquaculture personnel, the use of advanced management methods, and the implementation of aquaculture activities within the tolerance range of the ecological environment of the waters, which may finally realize the harmonious coexistence of aquaculture activities and the natural environment. The greening of aquaculture products is reflected in the final provision of safe and healthy high-quality aquatic products for consumers in aquaculture activities, which can increase profits while meeting market needs, help maintain motivation to continue to produce

high-quality aquatic products, and realize the harmonious development of people and aquaculture activities.

Therefore, this paper defines the green development of aquaculture as the process in which producers use advanced management concepts and green production methods, rationally allocate production factors and the efficient use of natural resources, and produce safe and high-quality aquatic products that meet market demand and increase farmers' income within the scope of water ecological environment capacity and resource-carrying capacity. To realize this production process, on the one hand, economic interests need to continuously stimulate the production enthusiasm of producers, and on the other hand, the fishery administration department needs to strictly supervise the production behavior of producers. In summary, this paper believes that the main connotation of the green development of the aquaculture industry should include four aspects: aquaculture resource input, technological innovation and promotion, administrative supervision, and efficiency development level.

### 2.2. Construction of the Evaluation Criteria System

Following the principles of scientificity, completeness, comparability, operability, and quantifiability of index design, this paper constructs an evaluation system of green development indicators for the aquaculture industry with 18 specific indicators, including 4 criteria layers of aquaculture resource input, technological innovation and promotion, administrative supervision, and efficiency development level.

Aquaculture is a resource-input industry, and aquaculture resources are the material basis for the green development of aquaculture. Resource input such as production factors is an important basis for determining the green development of the aquaculture industry, and the "Opinions on Accelerating the Green Development of the Aquaculture Industry" proposes to strengthen the scientific layout, stabilize the healthy aquaculture area of aquatic products, and ensure the aquaculture production space. Therefore, this paper selects the aquaculture area, number of aquaculture practitioners, and investment degree of aquaculture machinery per hectare to measure the input of aquaculture resources. The stable area of aquaculture waters directly determines the production and development of aquaculture. The sufficiency of labor resources can reflect the ability of an industry to absorb labor and reflect the level of industrial development. The input of machinery can reflect the current efficiency of resource utilization.

Aquaculture technology is an important support for the green development of aquaculture. On the one hand, the innovation and development of aquaculture technology can improve the technical level of the entire industry and solve a variety of problems faced in the current aquaculture process, such as seedlings, aquaculture disease prevention and control, aquaculture water environmental control, and other problems; on the other hand, technological progress can reduce breeding costs through the transformation of achievements, promote the improvement of breeding efficiency, improve breeding quality, and thus improve economic benefits. The promotion of aquaculture technology is equally important because the promotion of aquaculture technology can make up for the low education level of farmers, improve the awareness of safe and green production, and promote the green development of aquaculture. Therefore, this paper starts by describing the technological innovation ability and technology promotion efficiency to reflect the level of aquaculture technology, selects the per capita number of technical achievements and the per capita number of scientific and technological papers published by fishery technology departments to measure their technological innovation ability, selects the proportion of aquatic technology extension personnel above intermediate level and the number of aquatic technology extension institutions to measure the efficiency of technology promotion, and selects aquatic technology promotion funds to measure the government's support for technology promotion.

Administrative supervision is an important component of the green development of aquaculture. By strengthening law enforcement supervision, fishery administration

departments increase the investigation and handling of illegal acts, crack down on the use of illegal drugs, and improve a series of aquatic product quality and safety supervision and management systems so that producers can strengthen the standardized use of fishery drugs, fishing bait, and industrial disinfectants in fishery aquaculture, create a good industry environment, protect the interests of law-abiding practitioners, and ensure the quality and safety of aquaculture products. Therefore, this paper selects the inspection and testing batches of aquatic products, the number of fishery law enforcement agencies, the number of fishery administration management personnel, and the proportion of fishery administration management personnel with a bachelor's degree or above to measure the strength of fishery administration supervision and the quality of fishery administration management personnel.

Efficient development is the fundamental driving force for the green development of aquaculture. Under the premise of social, economic, resource, and environmental sustainability, green development emphasizes people's subjective initiative, pays more attention to the greening of the process, and brings positive improvements to resources and the environment through changing human behavior. Aquaculture operators are rational economic people, and the production behaviors adopted are all for obtaining the greatest economic benefits at the lowest economic cost, so the economic benefits of farmers are the fundamental driving force for them to continuously improve production methods and choose green and high-quality production methods. The current level of production is the basis for its further development towards green production. Therefore, this paper selects the proportion of disaster-affected aquaculture areas caused by disease, the proportion of aquatic product loss caused by disease, the output value per unit aquaculture area, and the output per unit aquaculture area to reflect the current production level and selects the per capita disposable income and per capita net income of fisheries to reflect the economic benefits of farmers.

The evaluation system of aquaculture green development indicators is shown in Table 1:

**Table 1.** Evaluation system for green development of aquaculture.

| Target Layer A | Criterion Layer Xi | Index Layer Xij |
|---|---|---|
| Evaluation of green development of aquaculture in China A | Aquaculture resource input X1 | Aquaculture area X11/hectare<br>Number of aquaculture practitioners X12/people<br>Investment degree of aquaculture machinery per hectare X13/kilowatt |
| | Technological innovation and promotion X2 | Number of technical outcomes per capita X21/number<br>All published scientific papers X22/piece<br>Proportion of aquaculture technology extension personnel at intermediate and above levels X23/%<br>Number of aquatic technology promotion institutions X24/number<br>Aquatic technology promotion fund X25/ten thousand yuan |
| | Administrative supervision X3 | Inspection testing batch X31/batch<br>Number of Fisheries enforcement agencies X32/number<br>Number of fishery management personnel X33/people<br>Proportion of fishery management personnel with bachelor's degree or above X34/% |
| | Efficient development level X4 | Proportion of disaster-affected aquaculture area caused by disease X41/%<br>Proportion of aquatic product loss caused by disease X42/%<br>Output value per unit aquaculture area X43/ten thousand yuan<br>Yield per aquaculture area X44/ton<br>Per capita disposable income X45/yuan<br>Per capita net income of fishery X46/yuan |

### 2.3. Entropy Weight Method

The entropy method is an objective weighting method that determines the index weight value according to the information provided by the observation value of each index. The numerical difference in the same index is positively related to the amount of information it contains. The greater the numerical difference, the more information it reflects, the greater the role of the index in the comprehensive evaluation, and the higher its weight [19–23]. The entropy method only depends on the degree of dispersion of the data itself, which can effectively avoid the impact of human or subjective factors on each evaluation index and make the evaluation results more objective and effective. The specific steps are as follows:

Step 1: indicators must be treated consistently before a comprehensive assessment is carried out to eliminate inconsistencies in terms of the magnitude and dimension of different measures. Indicators are divided into positive and negative indicators; positive indicators mean that the larger the value of the indicator, the better, while negative indicators are the opposite. The above indicators are positive indicators except for the proportion of disaster-affected aquaculture area caused by disease and the proportion of aquatic product loss caused by disease. There are different standardized treatments for positive and negative indicators, and the formula is as follows:

$$\text{Standardization of positive indicators}: \ X'_{\theta i} = \frac{X_{\theta i} - X_{min}}{X_{max} - X_{min}} \tag{1}$$

$$\text{Standardization of negative indicators}: \ X'_{\theta i} = \frac{X_{max} - X_{\theta i}}{X_{max} - X_{min}} \tag{2}$$

In the above formula, $X'_{\theta i}$ is the standardized value of the $i$th index in the $\theta$th year, and $X_{max}$ and $X_{min}$ represent the maximum and minimum values, respectively. In order to avoid meaningless logarithmic calculation when calculating the entropy value, the standardized data are uniformly added with 0.0001 for translation processing [16]. The rest of the steps are as follows:

$$Y'_{\theta i} = X'_{\theta i} + 0.0001 \tag{3}$$

Step 2: calculate the characteristic proportion $P_{\theta i}$ of the index in each year

$$P_{\theta i} = \frac{Y'_{\theta i}}{\sum Y'_{\theta i}} \tag{4}$$

Step 3: calculate the information entropy $H_i$ of the $i$th index

$$H_i = -k \sum P_{\theta i} ln P_{\theta i}, \ k = \frac{1}{ln\theta} \tag{5}$$

Step 4: calculate the difference coefficient $G_i$ of index $i$

$$G_i = 1 - H_i \tag{6}$$

Step 5: calculate the weight of each index $W_i$

$$W_i = \frac{G_i}{\sum G_i} \tag{7}$$

Step 6: The comprehensive score $H_{\theta i}$ of the green development level of the aquaculture industry in each year is calculated, and the value is between 0 and 1. The closer the comprehensive score is to 1, the higher the level of green development; the closer the comprehensive score is to 0, the lower the level of green development.

$$H_{\theta i} = \sum_i \left( W_i \, Y'_{\theta i} \right) \tag{8}$$

### 2.4. Analysis of Green Development of Aquaculture in China

According to the evaluation indicators of the green development of aquaculture in Table 1, we can find the corresponding data from 2012 to 2021, as shown in Table 2. The data of all indexes in this paper are from China Fishery Statistical Yearbook and China Rural Statistical Yearbook. Some of the indicators are obtained through calculation. The missing data in the yearbook are supplemented by the moving average method.

**Table 2.** Original data of index layer.

| Index Layer Xij | Year | | | | | | | | | |
|---|---|---|---|---|---|---|---|---|---|---|
| | 2012 | 2013 | 2014 | 2015 | 2016 | 2017 | 2018 | 2019 | 2020 | 2021 |
| X11 | 8,088,403 | 8,321,699 | 8,386,360 | 8,465,004 | 8,346,339 | 7,449,034 | 7,189,524 | 7,108,497 | 7,036,106 | 7,009,377 |
| X12 | 5,214,333 | 5,191,739 | 5,124,211 | 5,103,175 | 5,021,686 | 4,901,871 | 4,742,727 | 4,663,678 | 4,575,402 | 4,353,995 |
| X13 | 0.3052 | 0.3000 | 0.2873 | 0.2820 | 0.2771 | 0.2819 | 0.2628 | 0.2182 | 0.2019 | 0.2023 |
| X21 | 0.0057 | 0.0060 | 0.0064 | 0.0071 | 0.0078 | 0.0055 | 0.0043 | 0.0043 | 0.0049 | 0.0061 |
| X22 | 0.0355 | 0.0361 | 0.0362 | 0.0381 | 0.0406 | 0.0427 | 0.0424 | 0.0472 | 0.0434 | 0.0489 |
| X23 | 0.3056 | 0.3233 | 0.3275 | 0.3347 | 0.3882 | 0.395 | 0.4215 | 0.4514 | 0.4588 | 0.4828 |
| X24 | 14711 | 14,728 | 14755 | 14398 | 13463 | 12305 | 11976 | 11,705 | 11373 | 10846 |
| X25 | 183,036.76 | 180,188.35 | 201,802.92 | 232,935.5 | 289,540.41 | 314,529.47 | 335,038.54 | 372,032.56 | 391,885.66 | 401,731.31 |
| X31 | 182,896.98 | 172,562.54 | 158,622.05 | 142,897.10 | 136,632 | 190,836 | 380,427 | 194,739 | 226,613 | 264,560 |
| X32 | 2969 | 2949 | 2949 | 3124 | 2780 | 2679 | 2633 | 2587 | 2609 | 2629 |
| X33 | 36040 | 35139 | 35139 | 36620 | 39,106 | 38,364 | 35,574 | 32,784 | 32,870 | 34,599 |
| X34 | 0.289 | 0.2867 | 0.2867 | 0.3022 | 0.3449 | 0.3504 | 0.3935 | 0.444 | 0.4441 | 0.4716 |
| X41 | 0.0218 | 0.0182 | 0.0205 | 0.018 | 0.0168 | 0.0214 | 0.0213 | 0.0209 | 0.0177 | 0.0178 |
| X42 | 0.0059 | 0.0055 | 0.0046 | 0.0047 | 0.0052 | 0.0042 | 0.0041 | 0.0034 | 0.0027 | 0.0027 |
| X43 | 7.9859 | 8.7362 | 9.4058 | 9.7753 | 10.7275 | 12.3286 | 13.1528 | 13.7327 | 14.5298 | 16.7996 |
| X44 | 5.3019 | 5.4576 | 5.6621 | 5.8333 | 5.7429 | 6.5861 | 6.9421 | 7.1451 | 7.4248 | 7.696 |
| X45 | 18,871.47 | 18,107.30 | 17,343.127 | 16,578.954 | 15,814.78 | 17,277.43 | 18,809.45 | 20,159.54 | 20,857.92 | 22,589.99 |
| X46 | 7549.11 | 8744.70 | 9675.25 | 10,458.97 | 11,396.65 | 31,912.93 | 13,194.69 | 14,101.7 | 14,780.97 | 15,655.46 |

Data source: China Fishery Statistics Yearbook and China Rural Statistical Yearbook.

Then, according to the entropy method to calculate the comprehensive evaluation index, we first standardize the data to eliminate the inconsistency of different measurement indexes in terms of magnitude and dimension. According to Formulas (1)–(3), we can obtain the translated standardized decision matrix, as shown in Table 3 below.

**Table 3.** Standardization of index layer data.

| Index Layer Xij | Year | | | | | | | | | |
|---|---|---|---|---|---|---|---|---|---|---|
| | 2012 | 2013 | 2014 | 2015 | 2016 | 2017 | 2018 | 2019 | 2020 | 2021 |
| X11 | 0.7414 | 0.9017 | 0.9461 | 1.0001 | 0.9186 | 0.3021 | 0.1239 | 0.0682 | 0.0185 | 0.0001 |
| X12 | 1.0001 | 0.9738 | 0.8953 | 0.8709 | 0.7762 | 0.6369 | 0.4519 | 0.3601 | 0.2574 | 0.0001 |
| X13 | 1.0001 | 0.9504 | 0.8269 | 0.7764 | 0.7284 | 0.7746 | 0.5897 | 0.1584 | 0.0001 | 0.0043 |
| X21 | 0.3968 | 0.4959 | 0.5959 | 0.8126 | 1.0001 | 0.3490 | 0.0064 | 0.0001 | 0.1846 | 0.5275 |
| X22 | 0.0001 | 0.0443 | 0.0498 | 0.1939 | 0.3767 | 0.5376 | 0.5140 | 0.8707 | 0.5860 | 1.0001 |
| X23 | 0.0001 | 0.1000 | 0.1237 | 0.1643 | 0.4662 | 0.5046 | 0.6542 | 0.8229 | 0.8647 | 1.0001 |
| X24 | 0.9888 | 0.9932 | 1.0001 | 0.9088 | 0.6696 | 0.3733 | 0.2892 | 0.2198 | 0.1349 | 0.0001 |
| X25 | 0.0130 | 0.0001 | 0.0977 | 0.2382 | 0.4937 | 0.6065 | 0.6991 | 0.8660 | 0.9557 | 1.0001 |
| X31 | 0.1899 | 0.1475 | 0.0903 | 0.0258 | 0.0001 | 0.2224 | 1.0001 | 0.2384 | 0.3692 | 0.5248 |
| X32 | 0.7115 | 0.6742 | 0.6742 | 1.0001 | 0.3595 | 0.1714 | 0.0858 | 0.0001 | 0.0411 | 0.0783 |
| X33 | 0.5151 | 0.3726 | 0.3726 | 0.6069 | 1.0001 | 0.8827 | 0.4414 | 0.0001 | 0.0137 | 0.2872 |
| X34 | 0.0125 | 0.0001 | 0.0001 | 0.0839 | 0.3149 | 0.3446 | 0.5777 | 0.8508 | 0.8514 | 1.0001 |
| X41 | 0.0001 | 0.7201 | 0.2601 | 0.7601 | 1.0001 | 0.0801 | 0.1001 | 0.1801 | 0.8201 | 0.8001 |
| X42 | 0.0001 | 0.1251 | 0.4064 | 0.3751 | 0.2189 | 0.5314 | 0.5626 | 0.7814 | 1.0001 | 1.0001 |
| X43 | 0.0001 | 0.0852 | 0.1612 | 0.2031 | 0.3112 | 0.4928 | 0.5863 | 0.6521 | 0.7426 | 1.0001 |

**Table 3.** *Cont.*

| Index Layer Xij | Year | | | | | | | | | |
|---|---|---|---|---|---|---|---|---|---|---|
| | **2012** | **2013** | **2014** | **2015** | **2016** | **2017** | **2018** | **2019** | **2020** | **2021** |
| X44 | 0.0001 | 0.0651 | 0.1506 | 0.2221 | 0.1843 | 0.5365 | 0.6852 | 0.7700 | 0.8868 | 1.0001 |
| X45 | 0.4513 | 0.3385 | 0.2257 | 0.1129 | 0.0001 | 0.2160 | 0.4421 | 0.6414 | 0.7445 | 1.0001 |
| X46 | 0.0001 | 0.0492 | 0.0874 | 0.1195 | 0.1580 | 1.0001 | 0.2318 | 0.2690 | 0.2969 | 0.3328 |

Then, we calculate the entropy weight according to Formulas (4)–(7) and obtain the following Table 4.

**Table 4.** Entropy weight of index layer.

| Criterion Layer Xi | Weight | Index Layer Xij | Weight |
|---|---|---|---|
| X1 | 0.1443 | X11 | 0.0681 |
| | | X12 | 0.0295 |
| | | X13 | 0.0468 |
| X2 | 0.2632 | X21 | 0.0504 |
| | | X22 | 0.0600 |
| | | X23 | 0.0510 |
| | | X24 | 0.0445 |
| | | X25 | 0.0573 |
| X3 | 0.2734 | X31 | 0.0726 |
| | | X32 | 0.0713 |
| | | X33 | 0.0472 |
| | | X34 | 0.0823 |
| X4 | 0.3190 | X41 | 0.0565 |
| | | X42 | 0.0412 |
| | | X43 | 0.0486 |
| | | X44 | 0.0563 |
| | | X45 | 0.0438 |
| | | X46 | 0.0727 |

According to Formula (8), the comprehensive evaluation results of national aquaculture green development from 2012 to 2021 are obtained, as shown in Table 5 below and Figure 1.

**Table 5.** The evaluation results of comprehensive and four criteria.

| Year | A | X1 | X2 | X3 | X4 |
|---|---|---|---|---|---|
| 2012 | 0.3012 | 0.8780 | 0.2462 | 0.3287 | 0.0621 |
| 2013 | 0.3599 | 0.9322 | 0.2926 | 0.2794 | 0.2257 |
| 2014 | 0.3551 | 0.8971 | 0.3399 | 0.2642 | 0.2005 |
| 2015 | 0.4483 | 0.9012 | 0.4373 | 0.3977 | 0.2958 |
| 2016 | 0.4756 | 0.8279 | 0.5885 | 0.3612 | 0.3211 |
| 2017 | 0.4637 | 0.5236 | 0.4823 | 0.3599 | 0.5101 |
| 2018 | 0.4460 | 0.3418 | 0.4462 | 0.5381 | 0.4140 |
| 2019 | 0.4286 | 0.1570 | 0.5835 | 0.3194 | 0.5172 |
| 2020 | 0.4863 | 0.0613 | 0.5672 | 0.3673 | 0.7136 |
| 2021 | 0.5939 | 0.0014 | 0.7403 | 0.5104 | 0.8126 |

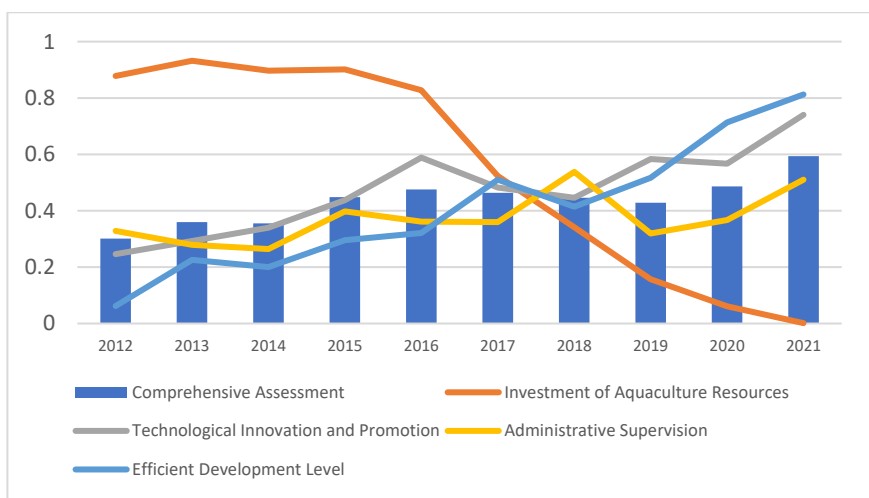

**Figure 1.** Trend of assessment results of green development of aquaculture from 2012 to 2021.

From the analysis of assessment indicators, the smaller the entropy value, the greater the entropy weight, which indicates that the indicator is more important than other indicators and contains more information, which should be the focus of attention. From the four criteria layers shown in Table 4, it can be seen that the weight coefficient of efficient development level (0.3190) is the highest, the weight coefficients of administrative supervision (0.2734) and technological innovation and promotion (0.2632) are slightly lower, and the weight coefficient of aquaculture resource input (0.1443) is the lowest, which shows that the level of aquaculture production and the living standard of farmers have the greatest impact on the green development of aquaculture, while the impact of technological innovation and promotion is slightly smaller and the input of aquaculture resources has the least impact on it. From further analysis of the 18 selected indicators, the proportion of fishery administration managers with a bachelor's degree or above, the per capita net income of fisheries, and the inspection and testing batch were found to be the three influencing factors with the highest weight coefficients, which indicates that in the process of greening aquaculture in China, the rigid constraints of administrative supervision and the economic benefits brought by green aquaculture have a greater impact on the green development of aquaculture and should be focused on in future development.

From the comprehensive assessment results, we can see the green development status and trend of aquaculture in China. From Table 5 and Figure 1, we can see that the overall green development of aquaculture in China shows a fluctuating upward trend, reaching the highest level in 2021, and the level of green development is significantly higher than that of ten years ago, with the growth rate reaching 97.18%.

In addition, technological innovation and promotion, administrative supervision, and efficient development level all showed a fluctuating upward trend, and the aquaculture input showed a downward trend. From the data of aquaculture input indicators, it can be seen that its aquaculture area and labor force have decreased to varying degrees in recent years, and from 2012 to 2021, the aquaculture area has shrunk from 8,088,403 hectares to 7,009,377 hectares, a decrease of 13.34%; the number of workers decreased from 5,214,333 to 4,353,995, a decrease of 16.5%. It can be seen that the availability and quality of water and the decrease in the aquaculture population are important reasons for the decline in the indicator layer of aquaculture inputs. Previous studies have shown that the early growth of aquaculture in China was achieved through the unsustainable development of many aquatic resources, which led to ecosystem degradation and habitat and biodiversity loss [24,25]. Aquaculture itself has become one of the main factors leading to the degradation of China's environment and ecosystem. At the same time, with the remarkable development of China's economy, rapid urbanization and industrialization have introduced external pollution sources such as synthetic and organic pollutants into

the aquaculture ecosystem. Growth industries other than aquaculture are competing with aquaculture for coastal and offshore waters [26]. In addition, the rapid development of urbanization and the improvement of people's education level have also led to a gradual decrease in the population of breeding labor. This means that it is not feasible to increase production by expanding the aquaculture area and increasing labor input. In the long run, strengthening technological innovation and improving technical efficiency may be important solutions for the green and sustainable development of aquaculture in China and even the world.

### 2.5. Suggestions

According to the research results, from the perspective of the weight of the criterion layer, the level of efficient development and administrative supervision have a greater impact on the green development of aquaculture in China. From the perspective of the weight of the index layer, the proportion of fishery administration managers with a bachelor's degree or above, the per capita net income of fisheries, and the inspection and testing batch are the three influencing factors with the highest weight coefficients. From the perspective of development trends, the input of aquaculture resources is a factor inhibiting the green development of aquaculture.

Based on the above research conclusions, this paper puts forward relevant suggestions to promote the efficient development of aquaculture, increase the economic income of producers, improve the level of fishery management supervision services, and make full use of the comparative advantages of each region.

Firstly, our recommendations are to strengthen aquaculture technology innovation and promotion to help the efficient development of aquaculture. We must attach importance to aquaculture technology innovation, strengthen the level of fishery scientific and technological innovation and the transformation of technological achievements, and carry out technology promotion and training activities in innovative forms. We can also increase investment in scientific research and technological research and development, build a scientific and technological exchange platform for the aquaculture industry at the national level, encourage the free flow of technical elements across provinces, strengthen scientific and technological exchanges, share innovative results, and enhance the ability to resist diseases and natural disasters. Additional suggestions are to strengthen the construction of technical promotion teams, introduce high-quality talents in related specialties, and improve the professionalism and overall quality of the team. It is also recommended to increase funding investment in grassroots aquatic technology extension units, encourage fishery technology extension departments to cooperate with scientific research institutions in colleges and universities, combine technical support for farmers with scientific research and teaching practice, implement technical training, effectively help producers transform green production methods, and improve farmers' green production capacity.

Secondly, it is recommended to improve subsidy policies, build a supply and demand market information platform, and increase the economic income of farmers. Relevant government departments should increase subsidy support for green aquaculture, improve the subsidy policy for the green aquaculture development model, and give incentives and subsidies such as upgrading tailwater treatment facilities, including circulating water treatment facilities, ecological ditches, ecological ponds, and undercurrent wetlands, so as to reduce the opportunity cost of producers switching to green production methods. Furthermore, it is suggested to build an information exchange service platform, smooth the flow of information between market demand and production and supply, strengthen the adaptability between the consumption end and the production end of aquatic products, promote the formation of a market mechanism with high quality and high price, and continuously increase the income of aquaculture subjects.

Thirdly, fishery administration departments should make full use of their supervision and service functions and strengthen the supervision and service level of aquaculture production. They should further improve the quarantine of aquatic seed production

areas, strengthen the punishment of illegal use of fishery drugs, increase the number of random inspections and testing batches of aquatic products in production areas, make full use of supervision functions, ensure the green safety of aquaculture processes and aquatic products, and standardize the market environment. Additionally, efforts can be made to strengthen the construction of fishery administration law enforcement teams and improve the law enforcement and service capacity of fishery administration teams. At the same time, the Ministry of Fisheries should also speed up the preparation of rules and standards to provide policy support for the green development of the aquaculture industry. Relevant government departments should, in accordance with the law, prepare a tidal planning system for aquaculture waters, strengthen the protection and management of fishery resources, and develop integrated rice and fish breeding and factory aquaculture in accordance with local conditions to ensure aquaculture production spaces.

Finally, according to local conditions, we should make full use of the advantages of regional characteristics and increase the investment in green production in the main fishery production areas. The main fishery production areas have the advantages of a vast aquaculture water surface, long aquaculture history, rich labor force, and other resources. The development of green aquaculture in the main fishery production areas makes it easy to produce scale effects. The green development process of aquaculture can be accelerated by strengthening the construction of fishery infrastructure, standardizing aquaculture ponds, water purification, and drainage systems, and increasing capital investment. Each province can start from the cultivation of high-quality aquatic seeds, aquaculture technology research, and the construction of demonstration bases for characteristic green aquaculture models according to their own actual conditions and making use of the advantages that have been formed in a targeted manner to drive the overall development of China's aquaculture industry.

## 3. Conclusions

The green development of aquaculture can promote the improvement of the ecological environment and aquatic product quality in aquaculture waters. The evaluation of the green development of aquaculture is of great significance for comprehensively grasping the current situation of green development of the aquaculture industry and providing a basis for policy formulation. The results showed that the overall green development of the aquaculture industry in China was good, showing a fluctuating upward trend, with the level of efficient development (0.3190) having the greatest impact on it, administrative supervision (0.2734) and technological innovation and promotion (0.2632) slightly weakening the impact, and the input of aquaculture resources (0.1443) having the least impact. The proportion of fishery administration management personnel with a bachelor's degree or above, fishery per capita net income, and inspection and testing batches are the three most weighted influencing factors, which shows that in the process of aquaculture greening in China, the rigid constraints of administrative supervision and the economic benefits brought by green aquaculture have a greater impact on the green development of aquaculture and should be focused on in future development. The overall green development of China's aquaculture industry showed a fluctuating upward trend, reaching the highest level in 2021, and the green development level was greatly improved compared with ten years ago, with a growth rate of 97.18%. Technological innovation and promotion, administrative supervision, and efficient development level all showed a fluctuating upward trend, and the aquaculture input showed a downward trend, mainly due to the reduction in aquaculture areas and the reduction in aquaculture labor. Finally, countermeasures and suggestions are put forward from the aspects of improving relevant policies, improving supervision and service levels, strengthening scientific and technological innovation and the transformation of scientific and technological achievements, and making use of characteristic advantages according to local conditions so as to promote the green development of China's aquaculture industry. This paper comprehensively presents the current status

of green development of the aquaculture industry in China and provides a reference for later research.

Although the evaluation index system of aquaculture green development established in this paper has been thoroughly studied and refined, it still needs to be improved in many aspects. In addition, subject to the availability of indicator data, the scientific validity and authority of the selected values of some evaluation indicators have yet to be determined, which needs further in-depth study in later work. In addition, with the deepening of the research on green aquaculture and the emergence of new ideas and methods, the value standards of some indicators determined in this paper can be further studied in the future.

**Author Contributions:** Conceptualization, X.Y. and P.Y.; methodology, X.Y.; software, X.Y.; validation, X.Y. and P.Y.; formal analysis, X.Y.; investigation, X.Y.; resources, X.Y.; data curation, X.Y.; writing—original draft preparation, X.Y.; writing—review and editing, X.Y.; supervision, X.Y.; project administration, X.Y. All authors have read and agreed to the published version of the manuscript.

**Funding:** This research received no external funding.

**Institutional Review Board Statement:** Not applicable.

**Informed Consent Statement:** Not applicable.

**Data Availability Statement:** The data in this article are all from the China Fishery Statistics Yearbook.

**Conflicts of Interest:** The authors declare no conflict of interest. The funders had no role in the design of the study; in the collection, analyses, or interpretation of data; in the writing of the manuscript, or in the decision to publish the results.

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
