# Peer review of "Analysis of Green Development of Aquaculture in China Based on Entropy Method"

_sustainability, doi:10.3390/su15065585_

Round 1
Reviewer 1 Report
I think the topic of the paper is important, and I enjoyed reading it. However, there are points that need to be addressed.
1. It would be better to justify why certain indicators are chosen for the scoring system. First of all, it is difficult to understand how the author(s) defines "green development". Green development could be anything. The author(s) need to explain why these indicators are selected from the definition.
2. The selection of these indicators leads to another question. These indicators are quite different. It is unclear what the combined number based on these different indicators means, and therefore, the authors need to explain what the number could mean.
For example, let us consider the case of automobiles, we have two indicators: Speed and fuel. We can create a new index from these two different indicators. But when we calculate it, the number does not mean anything. Speed and fuel are very important indicators of driving a car, but the combined number does not tell any information. It seems that there is the same kind of problem here.
The justification for the entropy weight is weak. In line 97 (p2), the paper suddenly declares that this paper uses an entropy weight model without explaining why. The reason is given in line 136 (p3), but the justification is very weak. Section 2.2 describes the method but does not give us the justification.
Author Response
Dear Professor:
Thank you very much for your valuable advice on my thesis, I have benefited a lot. On this basis, I have made changes in turn according to your revisions, and the specific revisions are attached to the document, please refer to it. Thank you again.
Best wishes to you!

Reviewer 2 Report
1. Please check some sentences. There are also hyphens in some sentences unnecessarily.
2. Is there any references show any good result of using Entropy method? Please cite, if it is available.
3. Page 2, please check the citation style.
4. As topic No.2 is Materials and Methods, please separate a part of result and discussion.
5. The authors mention in 2.1 that four criteria and sub-criteria are selected, what are the supporting reasons to select them? Is there any academic or practical reasons? It would be great for the readers if they know the idea so later, they can use as a guidelines.
6. Line 165; "As shown in Table 1" seems not to be completed.
7. For equations 1 and 2, do the authors know in advance if a Standardization is negative or positive? Can you please provide more idea how to select equations 1 or 2 to be used?
8. As shown in eq.(5), a term from this eq should be Hj, however, the one in eq.6 is Hi, how are they related? Are they meaning criteria and sub-criteria? Will Hj = Hi? Can you please also identify Theta?
9. Line 205; the sentence seems not to be completed.
10. Please rearrange Tables 2 and 3. It is not easy to understand this way of separation.
11. There are 2 citations for Table 2 in the content, However, there is only 1 source below the table. Please check if an another one is important.
12. Can the authors please specify more idea on using data from 2012 - 2021. As Covid-19 pandemic is among these years, is there any effect on the results?
13. Table 5; What is A on the first row?
14. Table 3; Please add thee unit of each parameters.
15. From Figure 1 and Table 5, X1 weight seems to decrease to 0. Is there any specific reason on it?
16. As concluded in Line 283-287, are they compatible with the results of weight shown in Figure 1?
17. For a part of suggestions, it would help the readers more if the readers can understand the method of changing/comparing the weight numbers to the policy or suggestions that the authors provide.
18. It would be grateful if the authors could put the contents in Topic 3 and classify to the table or in any form that would be easier to follow.
19. Some part of Conclusions is repeatedly written. Please define only the conclusions.
20. I am not sure if Prof.Ping Ying, mentioned in Acknowledgements, is the same person as the corresponding author. If so, please rewrite/remove.
21. Please also check the references part, the writing styles are not correct.
Author Response

(The authors gave the same response as above.)

Reviewer 3 Report
Overall, this is a clear and well-written manuscript. The selected criteria and indicators used in the analysis of green development of aquaculture are suitable for evaluation. The results of analysis are well presented. The methodology is well written and described.

Author Response

(The authors gave the same response as above.)

Round 2
Reviewer 1 Report
Although I disagree with some points in the authors' response on the index, the present paper is OK for publication.